# Automated Rapid Estimation of Flood Depth using Digital Elevation Model and EOS-04 Satellite derived flood inundation

Lakshmi Amani Chimata[1], Suresh Babu Anuvala Setty Venkata[1], Shashi Vardhan Reddy Patlolla[1], Durga Rao Korada Hari Venkata[1], Sreenivas Kandrika[1], Prakash Chauhan[1]

[1]National Remote Sensing Centre (NRSC), Indian Space Research Organization (ISRO), Hyderabad, India

*Correspondence to*: Lakshmi Amani Chimata (amanichimata@gmail.com)

**Abstract.** Rapid and accurate flood assessment is crucial for effective disaster response, rehabilitation, and mitigation strategies. This study presents a fully automated framework for floodwater delineation and depth estimation using EOS-04 (RISAT-1A) Synthetic Aperture Radar (SAR) imagery and a Digital Elevation Model (DEM). This is the first study to apply

the established Automatic Tile-Based Segmentation method and the Height above the Nearest Drainage (HAND) tool to EOS-04 data for flood extent delineation. For flood depth estimation, this study introduces a novel application of the Trend Surface Analysis (TSA) technique, enabling rapid and data-efficient assessment. Unlike traditional hydrodynamic models that demand extensive datasets and computational resources, TSA operates using only the inundated water layer and DEM, providing a highly data efficient solution.

The methodology is applied to flood-prone regions in Andhra Pradesh, Assam, Bihar, and Uttar Pradesh, India. Validation of flood extent against optical data demonstrates accuracy greater than 90%. Flood depth estimation using TSA is validated by comparing water depths derived from river gauge stations with real-time field measurements and results from the Floodwater Depth Estimation Tool (FwDET). The TSA method achieves a root mean square error (RMSE) of 0.805, significantly outperforming FwDET's RMSE of 5.23. This integration of high-resolution SAR imagery and DEM represents a

transformative, automated solution for real-time flood monitoring and depth estimation, enhancing disaster management capabilities.

**Key terms:** Automation, Flood inundation, Flood depth

## 1 Introduction

Floods are frequent natural disasters that can have devastating consequences, including loss of life, destruction of property, and disruption of livelihoods. According to the National Disaster Management Authority (NDMA), India is highly susceptible to floods, with over 40 million hectares out of a total geographical area of 329 million hectares prone to flooding (https://ndma.gov.in/Natural-Hazards/Floods). A satellite-derived flood-affected area atlas (1998-2022) indicates that the flood-affected area in India is 15.8 million hectares, reflecting the impact of significant flood events and cyclones (https://ndma.gov.in/flood-hazard-atlases). However, satellite data may have limitations in capturing other flood-affected regions, such as flash floods of short duration and areas lacking satellite coverage during the flooding period. Certain rivers are critical, including the Brahmaputra and Barak in Assam, the Kosi and Ganga in Bihar, the Ganga and Yamuna in Uttar Pradesh, and the Godavari in Andhra Pradesh. Cyclone-prone states such as Odisha, Andhra Pradesh, West Bengal, and Gujarat account for 10 million hectares of flood-affected areas, necessitating detailed hazard zonation maps. This highlights the critical need for real-time flood mapping and monitoring, the implementation of automated flood mapping techniques, and the generation of accurate spatial flood depth information to support disaster management efforts in these regions.

Satellite data and flood inundation information are widely used for near real-time mapping and monitoring of flood events (Rizwan Sadiq et al., 2022). Accuracy in flood extent and depth is essential for effective relief and rehabilitation efforts in the field. In this context, both Optical and Microwave satellite data sets are utilized, with the latter being more frequently used due to its advantage of satellite data acquisition under all weather conditions including rain, clouds, and sunlight, unlike sun-synchronous Optical satellite sensors (Felix Greifeneder et al., 2013). Therefore, space-borne Synthetic Aperture Radar (SAR) systems are preferred for flood monitoring. The techniques for satellite-derived flood inundation mapping, flood depth estimation, and various case studies are examined from the literature. EOS-04, the latest launch from ISRO, is designed to provide near real-time flood mapping and monitoring capabilities. Equipped with SAR sensors, it operates in both ascending and descending modes across coarse resolution mode (CRS), medium resolution mode (MRS) and fine resolution mode (FRS) configurations (A. V. Suresh Babu et al., 2024). The present research focuses on using newly launched EOS-04 satellite data to develop a methodology for automated, rapid estimation of flood inundation mapping and flood depth estimation using the Digital Elevation Model.

SAR data uses unique properties of water to detect water covered areas. Generally, low backscatter measurements are possible in calm, open water surfaces with SAR data (Schlaffer et al., 2014). This property of SAR images makes distinguishing water from surrounding surfaces more effective, even though visual interpretation helps flood mapping (Pierdicca et al. 2008). A literature survey revealed several articles on using SAR images for flood detection using various methods viz. (i) backscatter value-based thresholding (Boni et al., 2016, Chini et al., 2017, Greifeneder et al., 2014, Manjusree et al., 2012, Marti-Cardona et al., 2013, Martinis et al., 2015a, Martinis et al., 2013, Martinis et al., 2009, Twele et al., 2016), (ii) Interferometric coherence

calculation (Chini et al., 2019), (iii) region growing and active contour model (Giustarini et al., 2013, Li et al., 2014, Matgen et al., 2011, Tong et al., 2018), (iv) object-oriented classification (Horritt et al., 2001, Kuenzer et al., 2013b, Mason et al., 2010, Pulvirenti et al., 2011), (iv) fuzzy classification (Martinis et al., 2015a, Twele et al., 2016), and (vi) change detection (Bazi et al., 2005, Giustarini et al., 2013, Martinis et al., 2011, Schlaffer et al., 2015, Shen et al., 2018). Among these methods, thresholding-based methods have been most widely used in the literature in part because they are computationally less time-consuming and meanwhile could yield comparable accuracy to the more complex segmentation approaches (Gstaiger et al., 2012; Kuenzer et al., 2013b). Among backscatter histogram thresholding algorithms, the Otsu method has been widely applied in image segmentation (Otsu 1979; Kittler and Illingworth 1986). This method can automatically calculate the global threshold based on the criterion of maximum between-class variance and has high classification accuracy for images with a uniform bimodal distribution of gray histogram. However, suppose the histogram is unimodal or has non-uniform illumination, the traditional Otsu algorithm will fail and favour the class with a significant variance to improve the classification accuracy (Xu, X et al., 2011; Yuan et al., 2015). If the object size is less than 10% of the whole area, the performance of Otsu degrades significantly, and it will not be helpful for water detection methods (Cao et al., 2019).

Francesca et al., (2007) have used the method of dividing the SAR image into an unsupervised split-based approach (SBA) for change detection. This method automatically splits the image into a set of non-overlapping sub-images of user-defined size. Then, the sub-images are sorted according to their probability of containing many changed pixels. Afterward, a subset of splits with a high likelihood of containing changes is selected and analysed. This same change detection technique is applied for flood detection by Bovolo and Bruzzone (2007) to identify tsunami-induced changes in multi-temporal imagery and Martinis (2015) for flood mapping TerraSAR-X data. In view of the above limitation in the Otsu method and with the merits of the change detection method, the present study introduced a novel approach combining the Otsu threshold method with a tile-based segmentation strategy for flood extent delineation in EOS-04 satellite.

However, there is a limitation to this technique when mapping in hilly areas. In very steep slopes, the hillside may appear completely dark, as no radar signal is returned at all, potentially leading to a false interpretation of water pixels. In addressing this issue, Giacomelli (1995) integrated a SAR image with a digital terrain model and employed a simple technique to exclude this false interpretation by utilizing slope, slope direction, and drainage information. Additionally, the Height Above the Nearest Drainage (HAND) tool has been used to exclude hilly areas, enhancing the efficiency of the extracted water layer output, as demonstrated by Nobre et al., (2011). In this approach, HAND raster values are appropriately classified to eliminate false interpretations in the water layer.

In addition to the availability of flood inundation information in near real-time, it is crucial to have access to spatial flood depth maps for directing rescue and relief operations, pooling necessary resources, determining road closures and accessibility, and conducting post-event analysis (Islam et al., 2001). Flood depth identification during or after flood events is critical for

assessing hazard levels and creating risk zone maps, which are essential for post-disaster flood mitigation planning. While direct surveying methods used to determine floodwater depth can be highly accurate, they are often influenced by weather conditions, are costly, and may require field crews to obtain authorization to access sensitive flooded areas. In light of this,

remote sensing-based techniques and digital elevation models (DEMs) are valuable for estimating flood depth (Ismail Elkhrachy et al., 2022). Various hydrodynamic models such as HEC-RAS, Delft-3D, and LISFLOOD-FP have been developed to simulate water levels and flood depths (Yalcin, 2018; Costabile et al., 2021). However, these models require extensive data inputs, such as rainfall, soil moisture, flood maps, gauge discharge, cross-sections, and other hydrological inputs, which result in significant computational time and resource requirements.


Cohen et al. (2007) developed a floodwater depth calculation model based on high-resolution flood extent and DEM layers, known as the FwDET (Floodwater Depth Estimation Tool). The FwDET model identifies the floodwater elevation for each cell within the flooded domain based on its nearest flood boundary grid cell. While FwDET has been evaluated as one of the most effective tools for estimating flood depth from remote sensing-derived water extent and DEM (Teng et al., 2022), it has

inherent limitations. One critical limitation is that FwDET's floodwater depth maps are not continuous, often showing sharp transitions in values, which leads to linear stripes across the flooded domain (Cohen et al., 2018). Trend surface analysis has long been used by geographers, geologists, and ecologists to fit surfaces to data recorded at sample points scattered across a two-dimensional sample space (Chorley et al., 1965). In this paper, flood depth is estimated using a novel application of Trend Surface Analysis, which utilizes only the inundated water layer and a Digital Elevation Model (DEM). This study introduces

a novel approach for an end-to-end fully automated framework for floodwater delineation and depth estimation, utilizing real-time EOS-04 (RISAT-1A) Synthetic Aperture Radar (SAR) imagery and a Digital Elevation Model (DEM).

## 2. Study Area

The research focused on four significantly flood-affected regions in India's river plains: the Godavari, Brahmaputra, Kosi, and Ganga River basins. Table 1 provides detailed characteristics of flood proneness in these regions, while Figure 1 illustrates a

location map and the input EOS-04 satellite images of the study areas.

**Table 1** Study area locations and its characteristics

| S.No | Location (Lat/Lon) -decimal degrees | State -Districts Covered, River Basin | Study Area (Sq. Km) | Characteristics of the study area |
|------|-------------------------------------|---------------------------------------|---------------------|-----------------------------------|
| 1 | 17.4008°N to 17.8592°N and 80.9720°E to 81.6582°E | Andhra Pradesh- Alluri Sitaram Raju district | 72km × 50km | Receives maximum rainfall during South West Monsoon. 84% of annual rainfall falls |

| | | | |
|---|---|---|---|
| | | | during the period starting in mid-June and ending by mid-October |
| 2 | 25.9885°N to 26.7132°N and 90.6755°E to 91.8661°E | Guwahati and Barpeta areas of Assam State | 120km x 80km | The Brahmaputra River, known as, the lifeline of Assam, is one of the largest rivers in the world in terms of discharge |
| 3 | 25.0975°N to 25.7142°N and 86.2874°E to 87.6618°E | Bhagalpur of Bihar State | 138km x 68km | Floods frequently occur in Bihar over the Kosi River basin, hence the Kosi River is known as the "Sorrow of Bihar". Floods are generally caused by the breach of embankment along the Kosi river owing to intense rainfall during the monsoon season |
| 4 | 27.0138°N to 27.6943°N and 79.1919°E to 80.1584°E | Farrukhabad area of Uttar Pradesh | 95km x 75km | Vast majority of the state lies within the Gangetic Plain. The weather is of tropical monsoon type |

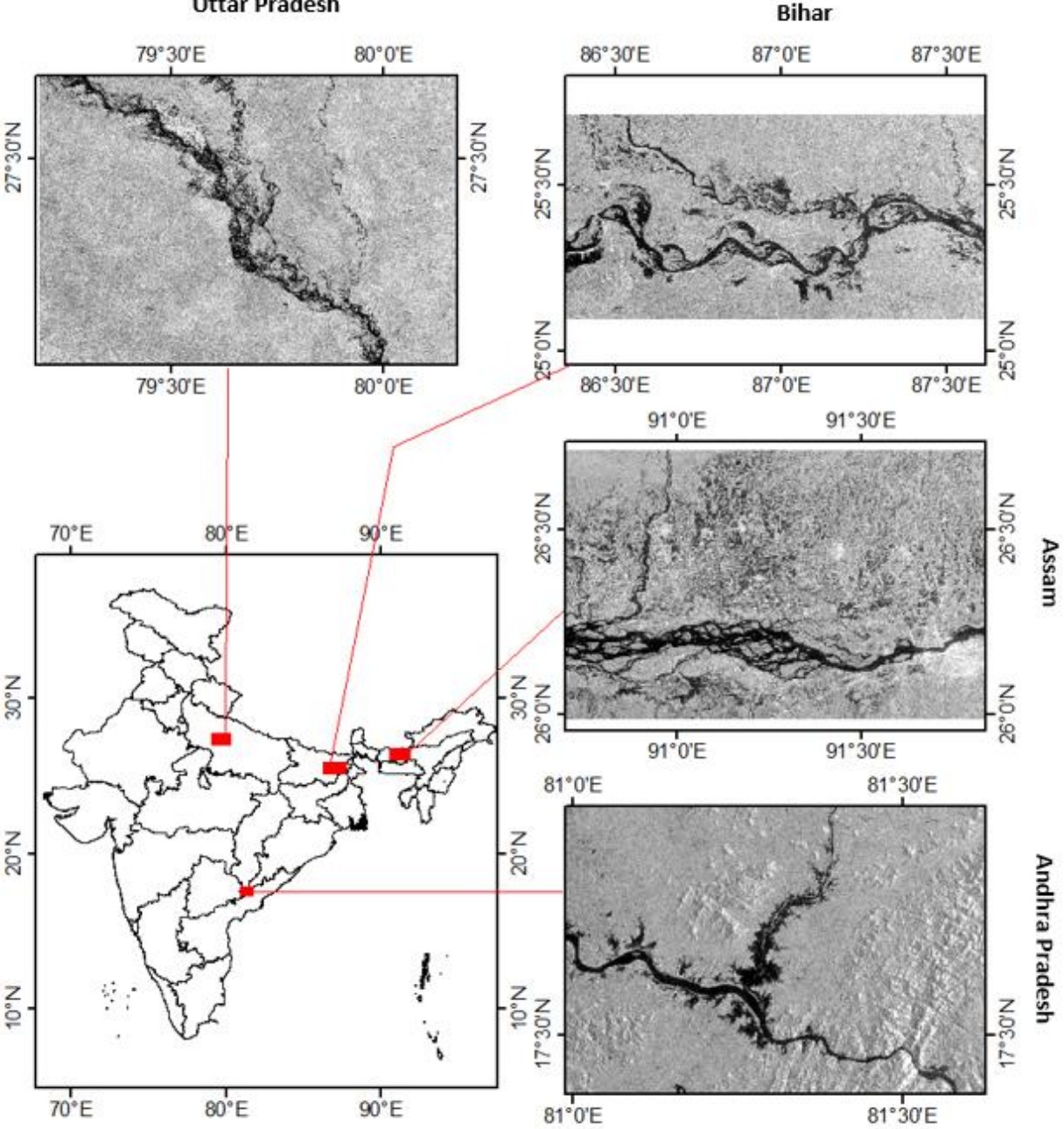

**Figure 1** Map showing four study area locations: Andhra Pradesh, Assam, Bihar and Uttar Pradesh

## 3. Data used

Comprehensive details on the satellite data and the associated Digital Elevation Model (DEM) utilized for estimating flood inundation and depth are provided in Table 2. To validate the flood extent layers, optical datasets were employed, with additional specifics outlined in Table 3. Figure 2 illustrates the locations of river gauge stations and the field-measured water levels provided by the Central Water Commission (CWC) of India. In this Figure 2, permanent water bodies within each study area are clearly highlighted in blue

**Table 2** Satellite data and respective DEMs used for the flood extent and depth estimation

| S.No | Study Area | Satellite Sensor | Satellite data Spatial Resolution(meters) | Satellite date and Time | DEM used for the study area | DEM spatial Resolution (meters) | Size of Study Area (Square Km) |
|---|---|---|---|---|---|---|---|
| 1. | Andhra Pradesh | EOS-04, CRS Mode | 36 | 28th July 2023 at 18:00 hrs | LiDAR DEM | 5 | 3300 |
| 2. | Assam | EOS-04, CRS Mode | 36 | 20th June 2023 at 18:00 hrs | FAB - DEM COPERNICUS | 30 | 9600 |
| 3. | Bihar | EOS-04, MRS Mode | 18 | 3rd September 2023 at 06:00 hrs | FAB (Forest and Buildings removed) DEM COPERNICUS | 30 | 9384 |
| 4. | Uttar Pradesh | EOS-04, MRS Mode | 18 | 15th August 2023 at 06:00 hrs | FAB-DEM COPERNICUS | 30 | 7600 |

**Table 3** Optical data used for validation of flood extent

| S.No | Study Area | Optical Dataset | Satellite data Spatial Resolution(meters) | Satellite date |
|---|---|---|---|---|
| 1. | Bihar | Resourcesat-2 LISS-4 sensor | 5.8m | 3rd September 2023 |
| 2. | Uttar Pradesh | Landsat-8 | 15m | 15th August 2023 |

### 3.1. Satellite Data and Digital elevation models

The Earth Observation Satellite-04 (EOS-04) is a synthetic aperture radar (SAR) satellite operating in the C-band frequency range of 5.4 GHz. Positioned in a sun-synchronous orbit at an altitude of 524.87 km, it offers various imaging modes, including Fine Resolution Strip Mode (FRS), Medium Resolution ScanSAR Mode (MRS), Coarse Resolution ScanSAR Mode (CRS), and High-Resolution Spotlight Mode (HRS). These modes allow the satellite to capture data with different levels of detail and coverage. The resolution capability of the EOS-04 satellite ranges from 1 m to 50 m, enabling data acquisition at various spatial

resolutions.

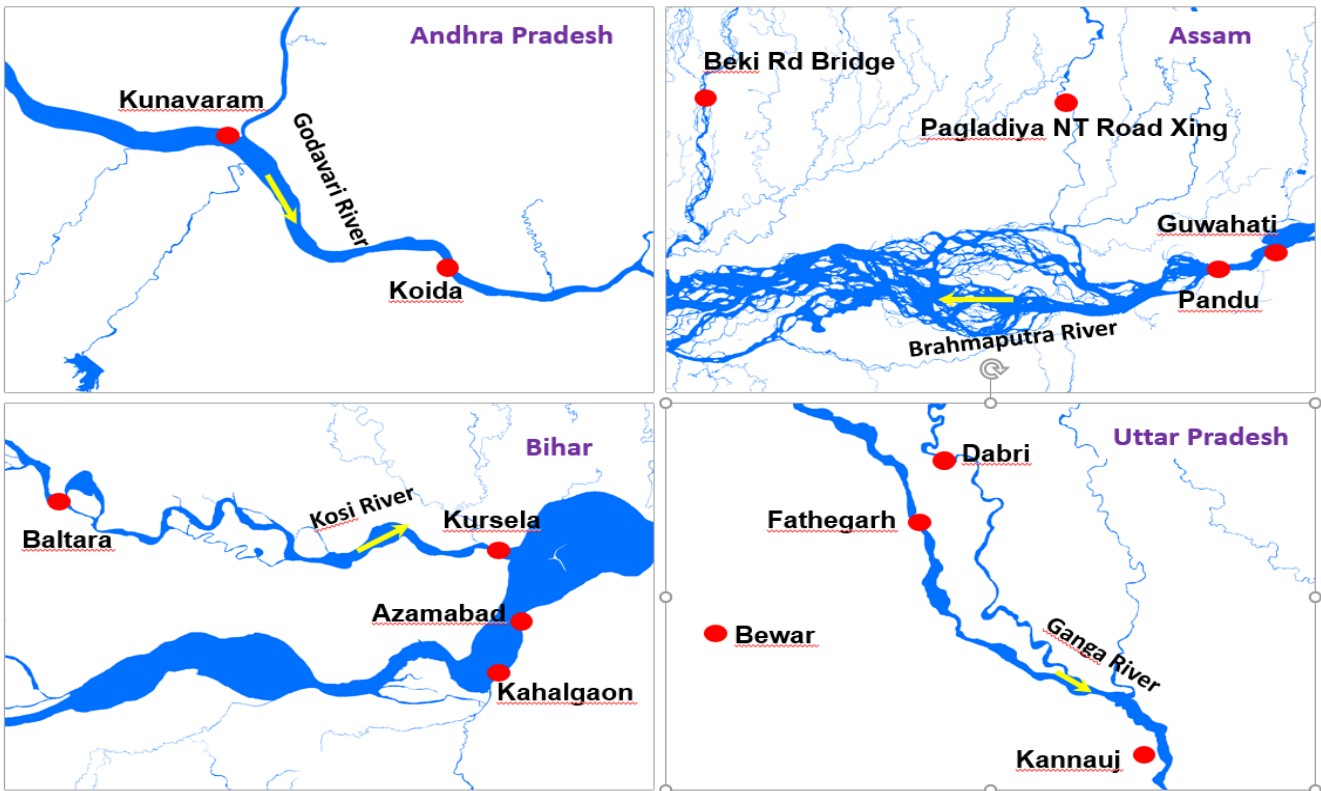

*Note:* *Blue colour represents permanent water bodies in each study area*

**Figure 2** River gauge station locations in Andhra Pradesh, Assam, Bihar and Uttar Pradesh.

**3.2. Field Measurements:**

Typically, water levels are measured using gauge stations installed along rivers. The Central Water Commission (CWC) of India provides hourly field measurements from these gauge stations, as illustrated in Figure 2, for various sites, and makes the information available on their website (https://ffs.india-water.gov.in/). Water levels recorded at the times corresponding to satellite acquisitions across all study areas are compared with the interpolated levels derived from the Trend Surface Analysis

(TSA). Table 3 presents the field-measured water levels from gauge stations corresponding to the specific dates and times of satellite acquisitions.

**Table 4** Field-measured water levels from gauge stations in the study area

| S.No | Study Area | Water Gauge Station Name | Field Measured Water Levels (meters) |
|------|-----------|--------------------------|--------------------------------------|
| 1 | Andhra Pradesh | Kunavaram | 41.02 |
| 2 | | Koida | 39.72 |
| 3 | Assam | Beki Rd Bridge | 44.92 |
| 4 | | Pangladiya NT Road Xing | 52.84 |
| 5 | | Pandu | 47.25 |
| 6 | | Guwahathi | 48.19 |
| 7 | Bihar | Baltara | 34.90 |
| 8 | | Kahalgaon | 31.08 |
| 9 | | Azamabad | 30.54 |
| 10 | | Kursela | 29.98 |
| 11 | Uttar Pradesh | Dabri | 137.18 |
| 12 | | Fathegarh | 137.78 |
| 13 | | Kannauj | 125.67 |
| 14 | | Bewar | 138.32 |

## 4. Methodology

The process of quickly estimating flood depth using the Digital Elevation Model and EOS-04 satellite involves several steps. These include generating a radar backscatter coefficient image from the raw satellite image, extracting the flood inundation layer using an automated tile-based segmentation method, obtaining terrain information prior to the flood event using a digital elevation model, interpolating floodwater surface levels through Trend Surface Analysis, and determining the spatial flood depth. The methodology is illustrated in the flow chart as shown in Figure 3(a) and Figure 3(b). A customized Python code

has been developed specifically for automated flood mapping and depth estimation using ArcGIS and GDAL libraries.

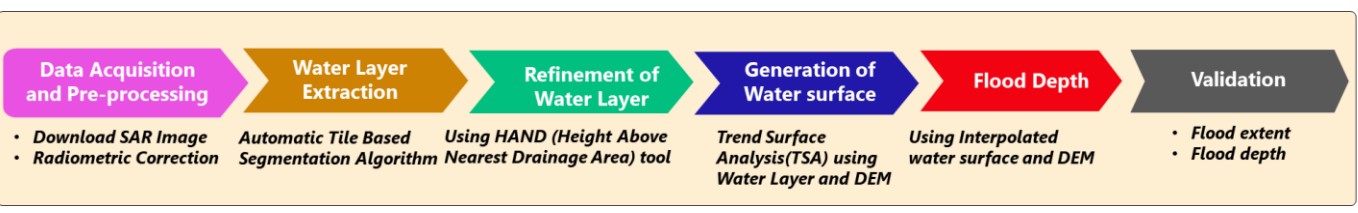

**(a)**

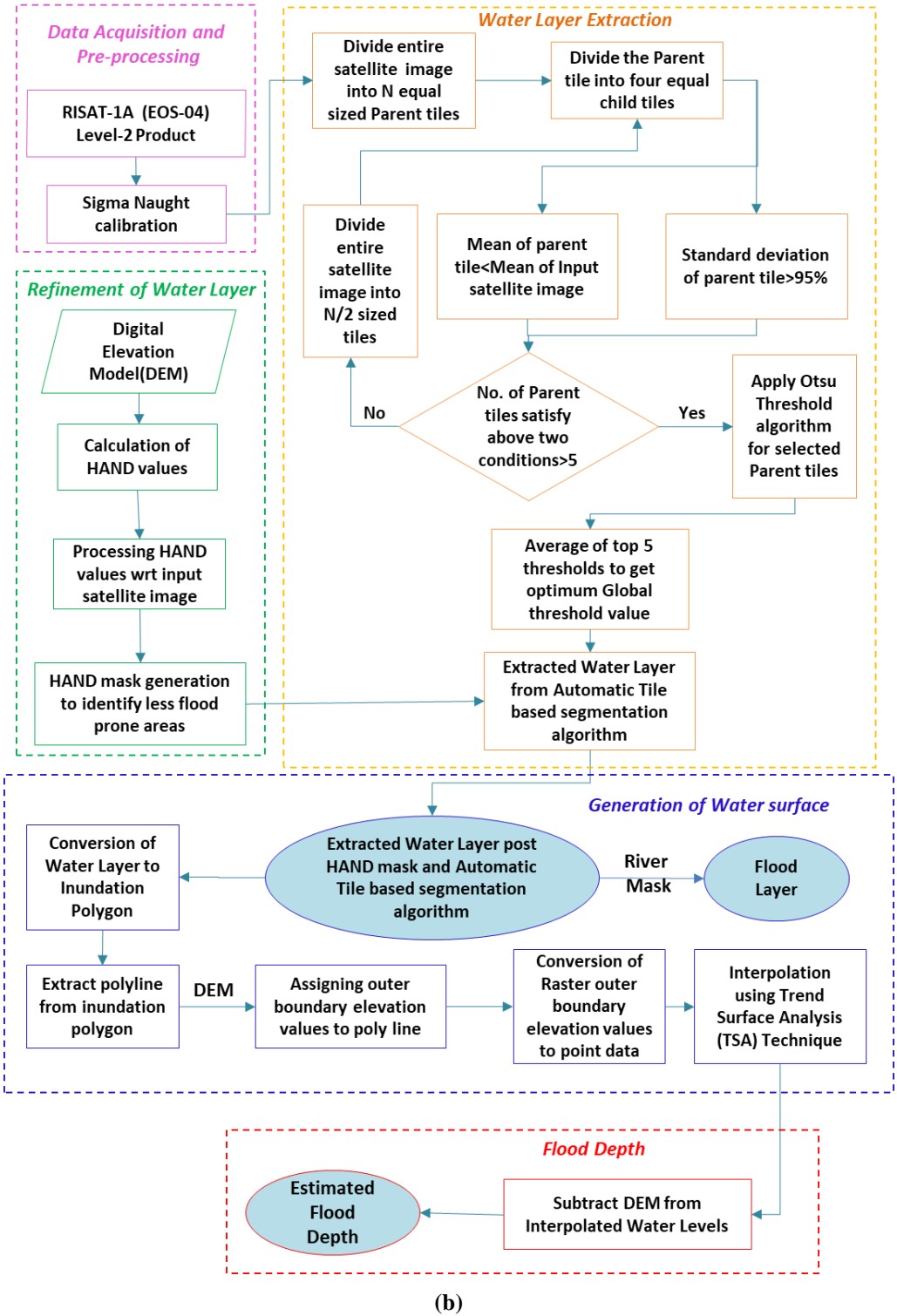

**Figure 3** Methodology overview (a) steps of the methodology (b) detailed flowchart illustrating the methodology.

## 4.1 Data Acquisition and Pre-processing

Any multi-sensor satellite data which is acquired past the flood event by satellite is hosted in the Indian Space Research Organisation (ISRO)'s Bhoonidhi portal and can be downloaded. Pre-processing of EOS-04 data involves both geometric and radiometric corrections before the application of data for flood extraction (A. V. Suresh Babu et al., 2024). Geometric correction ensures the spatial accuracy of the SAR data by aligning it with a coordinate system or correcting distortions caused by sensor geometry, Earth's curvature, and terrain variations. Radiometric correction involves adjusting the pixel values in the SAR data to accurately reflect the actual backscattered signal (Converts raw digital numbers (DNs) into physical quantities such as backscatter intensity) compensating for system and environmental effects. This ensures consistency across sensors and acquisition times. Radar backscatter represents the intensity of the radar signal reflected back to the sensor from the Earth's surface, providing valuable insights into surface roughness, moisture content, and material properties. By analysing radar backscatter, water bodies can be accurately identified, surface conditions can be properly assessed, and land and water classification can be improved in remote sensing applications. Radar backscatter coefficient values, i.e., Sigma Nought ($\sigma_o$), for the EOS-04 satellite image is computed as per the following equation (1):

$$\sigma_o(dB) = 20 * \log_{10}(DN) + 10 * \log_{10} \sin \theta_{inc} - CF \tag{1}$$

Where DN represents digital number (amplitude in Level-2 products), $\theta_{inc}$ is the per-pixel local incidence angle and CF is the Calibration Factor.

## 4.2 Water Layer Extraction

The water layer from the radiometrically calibrated image is extracted using the Automatic Tile-Based Segmentation Method, which involves partitioning the image into tiles, specific criteria-based tile selection, calculating thresholds, and classifying the image into water and non-water areas (Martinis et al., 2015) as illustrated in Figure 4. The image is partitioned into non-overlapping tiles of equal size (n x n pixels), referred to as parent tiles. If perfect partitioning is not possible, the last row and column tiles are adjusted to ensure they remain n x n pixels. Each parent tile is then subdivided into four equal-sized child tiles. For threshold calculation, tiles are selected based on two conditions:

1. The mean radar backscatter value of the parent tile should be lower than the mean backscatter value of the entire image, ensuring the tiles are located near the boundary between water and non-water areas.
2. The standard deviation of the parent tile must exceed 95% of the image's overall standard deviation, indicating significant variation within the tile, which enhances the classification of water and non-water areas.

If parent tiles that meet the specified conditions are less than 5% of the total tiles, the image is subdivided into smaller tiles (n/2 × n/2) and the standard deviation threshold is then relaxed to 90%, and the process is repeated until the selected tile is sufficient. Once the necessary tiles are chosen, all parent tiles that satisfy both conditions are processed using the Otsu thresholding technique. The global threshold value is calculated as the average of the individual thresholds from the selected

tiles and is used to classify the SAR image into water and non-water areas. This methodology is summarized in the flowchart presented in Figure 3(b).

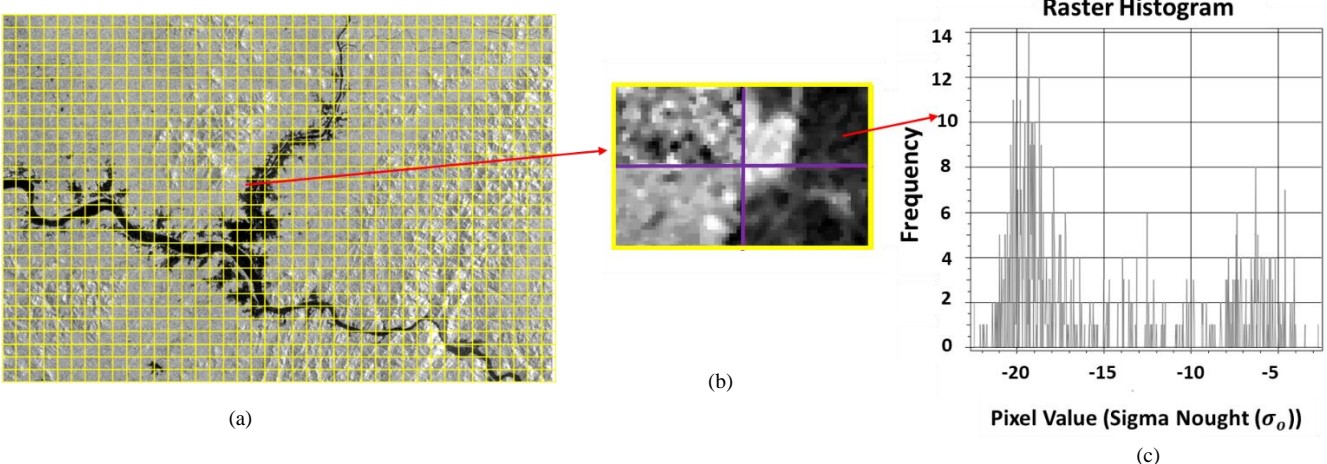

(a)

(b)

(c)

205

**Figure 4** Automatic Tile based segmentation of SAR image (a) division of SAR image into n parent tiles (b) division of parent tile to 4 child tiles (c) histogram of one child tile

### 4.3. Refinement of Water Layer

To improve the accuracy of water classification and eliminate false water areas such as shadows and stray pixels caused by speckle noise, the Height above Nearest Drainage (HAND) tool is employed. HAND is a terrain model that standardizes topography relative to the drainage network and characterizes local drainage potential. In a HAND raster, each pixel represents the vertical distance (in meters) from that point to the nearest drainage channel. The HAND tool facilitates the rapid identification of non-flooded areas by restricting the flood areas up to a HAND Value of 15 m following the local drainage direction toward the channel where water flows. According to Nobre et al. (2015), regions with HAND values greater than 15 meters are less vulnerable to flooding. Hence applying the HAND mask refines the water layer, significantly reducing misclassification and subsequently producing the flood layer by subtracting a permanent river mask. The creation of a HAND raster from a DEM involves several steps, illustrated in Figure 3(b). These steps include:

- Generating a seamless, hydrologically corrected DEM using the Fill tool.
- Defining flow paths using the Flow Direction function.
- Identifying the drainage network through Flow Accumulation.
- Calculating the HAND values using the D8 flow distance algorithm

## 4.4. Generation of Water Surface

To estimate flood depth, it is necessary to generate a water surface using only a 2D inundated water layer and a Digital Elevation Model (DEM) as input data using Trend Surface Analysis (TSA). TSA generated water surface offers a notable advantage over traditional hydrodynamic models, which are often data-intensive. This streamlined approach provides a simplified yet effective solution for flood depth estimation. Trend Surface Analysis uses global polynomial interpolation to generate a smooth surface defined by a mathematical function derived from input sample points. This technique effectively captures gradual changes and coarse-scale patterns in the data, producing a smooth surface that reflects the overall trend across the area of interest (Morton et al., 1974). TSA achieves this by fitting a polynomial function to known data points (outer boundary elevation points) and using this function to predict values at locations where data is unavailable inside the flood extent. In this study, the outer boundary elevation values derived from the DEM for the inundated water layer are used as input for the interpolation process. Based on the findings of Cohen et al. (2017), Huang et al. (2014), Brown et al. (2016), and Cian et al. (2018), it is assumed that the water surface in flooded areas is flat when calculating flood depth In natural flood scenarios, water does not flow in abrupt steps like it would over steep or terraced terrain. Instead, floodwaters tend to spread out smoothly, forming a surface with a gentle slope or remaining nearly flat across large areas. This occurs because floodwater follows the path of least resistance, gradually filling depressions and expanding outward rather than forming sharp elevation differences. Given this characteristic, using a first-degree polynomial equation in Trend Surface Analysis (TSA) is a rational approach for modelling the floodwater surface. A first-degree polynomial represents a linear trend, which effectively captures the gradual variation in water surface elevation across the flooded area. This ensures that the estimated water surface reflects the actual spread of floodwater rather than introducing artificial discontinuities that would arise if higher-degree polynomials or abrupt elevation changes were assumed. By implementing this method, the study enhances flood depth estimation accuracy, as it aligns with the natural behaviour of floodwaters. The approach provides a more realistic representation of inundation, ensuring that calculated flood depths are consistent with the actual hydrodynamic conditions observed during flooding events.

Mathematically, the observed elevation at any point along the outer boundary of the inundated water surface can be expressed as the sum of the predicted elevation from TSA and the residual error at that point:

$$Z_{observed} = f(x_i, y_i) + r_i$$

$Z_{observed}$ = The observed elevation value at the $i^{th}$ point inside the water surface

$f(x_i, y_i)$ = Polynomial function that predicts the elevation based on the coordinates $x_i$ (latitude) and $y_i$ (longitude).

$r_i$ = represents the residual at the $i^{th}$ point, which is the difference between the observed and predicted elevation.

The first-degree polynomial equation used in this study is defined as:

$$f(x_i, y_i) = ax_i + by_i + c$$

where a, b and c are constants that define the coefficients of the polynomial.

In real-world topographic surfaces, observed elevations rarely align perfectly with the predicted trend. Residuals $r_i$ quantify the discrepancy:

- A positive residual indicates that the observed elevation is above the trend surface.
- A negative residual indicates that the observed elevation is below the trend surface.

To determine the optimal coefficients a, b and c, the least squares criterion is employed, minimizing the sum of squared residuals (S):

$$S = \sum_{i=1}^{N} (r_i{}^2)$$

Where $S$ represents the sum of squared residuals and $r_i$ is the residual at the $i^{th}$ point

The process for deriving the TSA-interpolated surface is illustrated in Figure 5. First, the 2D water layer, obtained from the Automatic Tile based Segmentation method and the HAND tool, as shown in Figure 5(a), needs to be converted into polygon form. This layer is then transformed into a polygon that retains only the outer boundary segments. Next, the polygon is converted into raster format, and the corresponding outer boundary elevation values are assigned from the Digital Elevation Model (DEM). Since the TSA technique operates exclusively on point data, the raster is then converted to point form, as depicted in Figure 5(b). A first-degree polynomial surface is fit to the outer boundary elevation values.

Predicted elevations are computed across the inundated area, producing an interpolated TSA water surface as illustrated in Figure 5(c). The resulting TSA interpolated surface provides estimated water surface levels in meters,

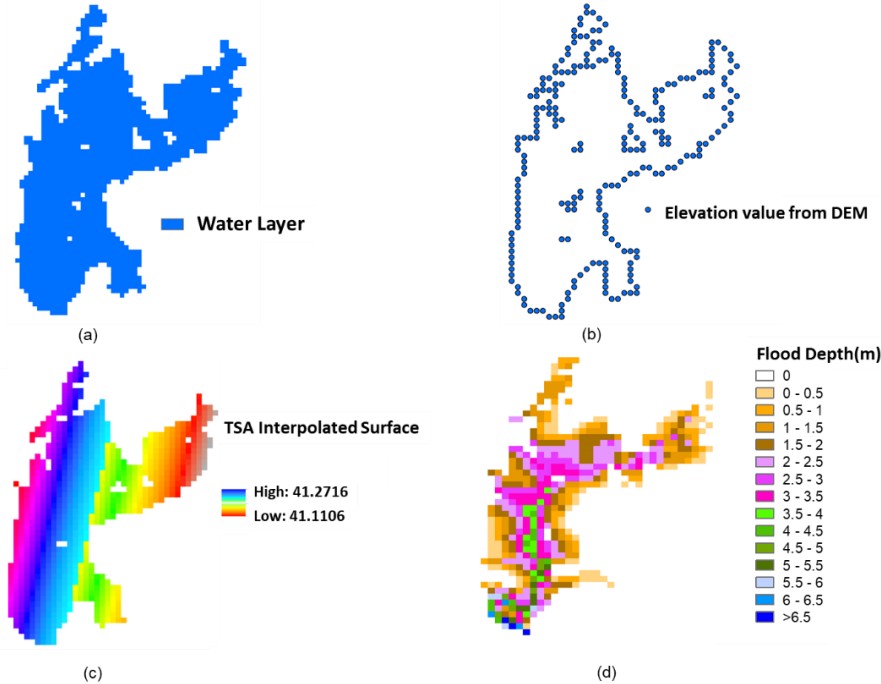

**Figure 5** Methodology for flood depth estimation using TSA technique: (a) water layer created using the Automatic Tile-based segmentation technique and HAND tool. (b) elevation values extracted for the outer boundary water layer from the Digital Elevation Model (DEM) as points. (c) the interpolated surface is generated using these elevation points through Trend Surface Analysis (TSA). (d) flood depth is estimated by subtracting DEM values from the interpolated water levels (above mean sea level).

## 4.5. Flood Depth

The calculation of flood depth is achieved by subtracting the Digital Elevation Model (DEM) from the water levels interpolated by the TSA. The resulting depth is expressed in meters, as depicted in Figure 5(d).

## 5. Results

This research focuses on automated rapid flood inundation and depth estimation using Synthetic Aperture Radar (SAR) imagery (EOS-04 data) integrating Automatic Tile-Based Segmentation Method and the Height above Nearest Drainage (HAND) tool is validated for flood extent against cloud-free optical satellite data for Bihar and Uttar Pradesh, as detailed in Section 5.1. Floodwater depth is estimated in all study areas and validated as per section 5.2. Accurate digital elevation models (DEMs) are critical for determining floodwater depth, but highly accurate DEMs are not available in all places, hence it is required to assess the sensitivity of flood depth against DEM characteristics. This study uses a high-resolution 5m LiDAR DEM and the 30m Copernicus FABDEM for the Godavari River to assess flood depth sensitivity, as described in Section 5.2.1. Flood depth estimation using Trend Surface Analysis (TSA) is conducted for the Godavari, Ganga, Brahmaputra, and Kosi rivers. The accuracy of TSA-derived flood depth estimates is assessed by comparing them with field-measured river water levels recorded by the Central Water Commission (CWC) for corresponding dates and times. Additionally, further comparisons with the Floodwater Depth Estimation Tool (FwDET) are comprehensively detailed in Section 5.2.2.

## 5.1. Flood Inundation Area Estimation and Validation

The flood inundation layer is created using the proposed method from EOS-04 data. Conducting fieldwork for flood map validation during a flood disaster is often challenging. Therefore, the accuracy of the delineated flood layer is assessed using a cloud-free optical satellite image from Landsat-8 of 15m resolution, which was acquired on the same date i.e., August 15, 2023 as the EOS-04 data of 18m in the Uttar Pradesh study area. Additionally, a Resourcesat-2 LISS-4 image of 5.8m, also obtained on the same date as EOS-04 i.e., September 3, 2023 was used for the Bihar study area. To extract water extent from the optical images, standard unsupervised classification techniques were applied using ERDAS Imagine software (Ali et al., 2017). The results of this analysis are presented in Figure 6 as shown below. Delineated flood pixels are shown in blue colour.

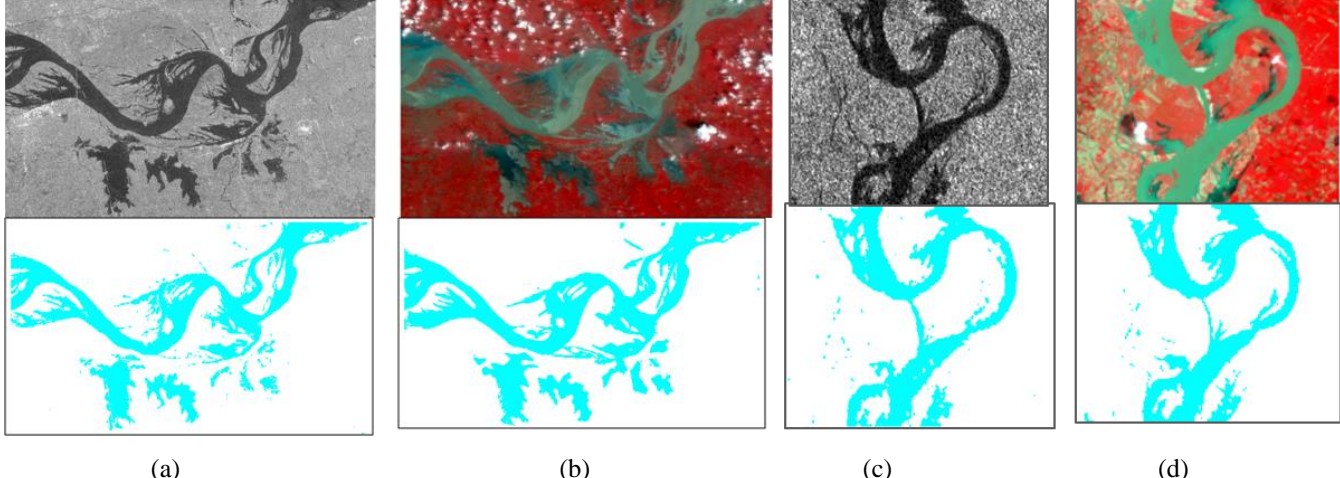

(a)       (b)       (c)       (d)

**Figure 6** Optical satellite data and EOS-04 data comparison. (a) EOS-04 data and the delineated flood layer using Automatic Tile-Based Segmentation Method and HAND tool in Bihar study area. (b) Resourcesat-2 LISS-4 data and the flood inundation layer were extracted using unsupervised classification in the Bihar study area. (c) EOS-04 data and the delineated flood layer using the Automatic Tile-Based Segmentation Method and HAND tool in the Uttar Pradesh study area. (d) LANDSAT-8 data

and the flood inundation layer extracted using unsupervised classification in the Uttar Pradesh study area

As a part of the accuracy test for flood extent using the proposed method, the Confusion matrix and performance metrics are computed for the Bihar and Uttar Pradesh study area with respect to respective optical datasets as detailed in Table 5 and Table 6.

**Table 5** Confusion matrix for flooded and non-flooded pixels in Bihar and Uttar Pradesh study areas

| Bihar | | |
|---|---|---|
| **Actual/Predicted** | **Flooded** | **Non-Flooded** |
| **Flooded** | 8,534,283 | 116,598 |
| **Non-Flooded** | 513,381 | 2,038,290 |

| Uttar Pradesh | | |
|---|---|---|
| **Actual/Predicted** | **Flooded** | **Non-Flooded** |
| **Flooded** | 174,506 | 6,391 |
| **Non-Flooded** | 11,577 | 37,686 |

**Table 6** Performance metrics for the flooded pixels in Bihar and Uttar Pradesh study areas

| **Study Area** | **Precision** | **Recall** | **F1-Score** | **Accuracy** |
|---|---|---|---|---|
| Bihar | 95% | 80% | 87% | **94%** |
| Uttar Pradesh | 86% | 76% | 81% | **92%** |

The flood delineation accuracy for the Bihar and Uttar Pradesh study areas exceeds 90% when compared to optical data as per Table 6. However, certain discrepancies are observed as per Table 5 due to the characteristics of microwave data. In shallow

flowing water areas, microwave sensors may incorrectly classify these regions as dry, unlike optical data, which accurately identifies the presence of water. Additionally, microwave data sometimes misinterprets moisture-laden areas as flooded, leading to overestimation.

Despite these limitations, the Automatic Tile-Based Segmentation Method combined with the HAND tool proves effective for generating flood maps rapidly using EOS-04 data. Since flood depth estimation relies on the delineated flood extent from SAR

images, this method offers a reliable approach for automatically detecting water layers, enabling efficient and accurate flood mapping in critical situations.

### 5.2. Floodwater Depth Estimation and Validation

### 5.2.1. Sensitivity of Flood Depth with DEM characteristics

The accuracy of floodwater depth measurements depends significantly on the accuracy and spatial resolution of the Digital

Elevation Model (DEM) as it plays a major role in the interpolation of floodwater depth. To assess this, an analysis was conducted in the Godavari flood plain area, utilizing two different DEM datasets. One DEM was derived from LiDAR data with a 5m spatial resolution and vertical accuracy of 15 cm, while the other was obtained from the public domain, specifically the Copernicus FABDEM, with an 8m vertical accuracy and 30m spatial resolution. This comparative study aims to evaluate the impact of public domain DEMs on the accuracy of floodwater depth estimation. Here, the flood depth is estimated in the

Godavari Flood plain study area using the Trend Surface Analysis (TSA) Technique. The results of this analysis are presented in the Figure 7 below. A scatter plot is drawn for comparison of flood depth values estimated using the TSA technique for LiDAR and Copernicus DEMs.

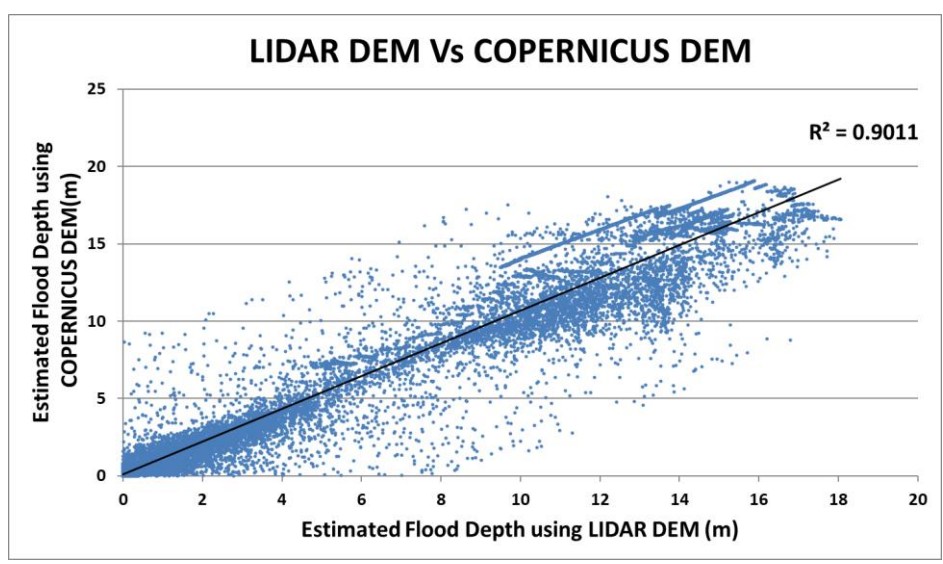

**Figure 7** Plot between LiDAR DEM and Copernicus DEM derived flood depths

The scatter plot above indicates that 90% of flood depth points derived from LiDAR and Copernicus DEMs match closely. The areas where discrepancies occur are predominantly in steep slope regions having elevation changes rapidly. Therefore, accurate LiDAR-derived DEMs are essential for estimating flood depths in steep areas. In contrast, for gentle slope areas, COPERNICUS DEM with a 30m spacing and vertical accuracy of 8m provides sufficiently accurate flood depth estimates as depths are relative heights.

### 5.2.2 Results of Flood Depth Estimation and Validation

The shape of flood layers varies across different areas, with some regions appearing wide, indicating a gentle slope, and others being narrow along rivers, suggesting a steeper gradient, as observed in the aforementioned case studies. There is an increasing demand for accurately determining flowing water surfaces to precisely estimate flood depths. Typically, the flowing water surface is derived through two steps: firstly, by collecting elevations along the flood inundation boundary, which represent varying heights of discrete points, and secondly, by fitting a surface across these elevation points using proposed interpolation methods.

### 5.2.2.1 Derivation of Flood depths using TSA technique in Study Areas

Given the dynamic nature of varying water levels of flowing rivers at different locations, employing trend surface analysis becomes essential for simulating the exact water surface, especially in large flooded areas with gentle slopes. This process involves calculating floodwater depths based on DEM resolution at specific locations, such as pixels. The Figure 8 below illustrates the flood depths in four areas of gentle slope. For the Andhra Pradesh study area, LiDAR DEM derived floodwater depth using TSA is illustrated in Figure 8(a). For the remaining three study areas such as Bihar, Assam and Uttar Pradesh, publicly available Copernicus DEM is used to estimate floodwater depth using the TSA technique.

From Figure 8, it is evident that the flood depth is greater in river areas and it is represented in blue colour. Flood depths derived from the TSA technique are smooth and continuous.

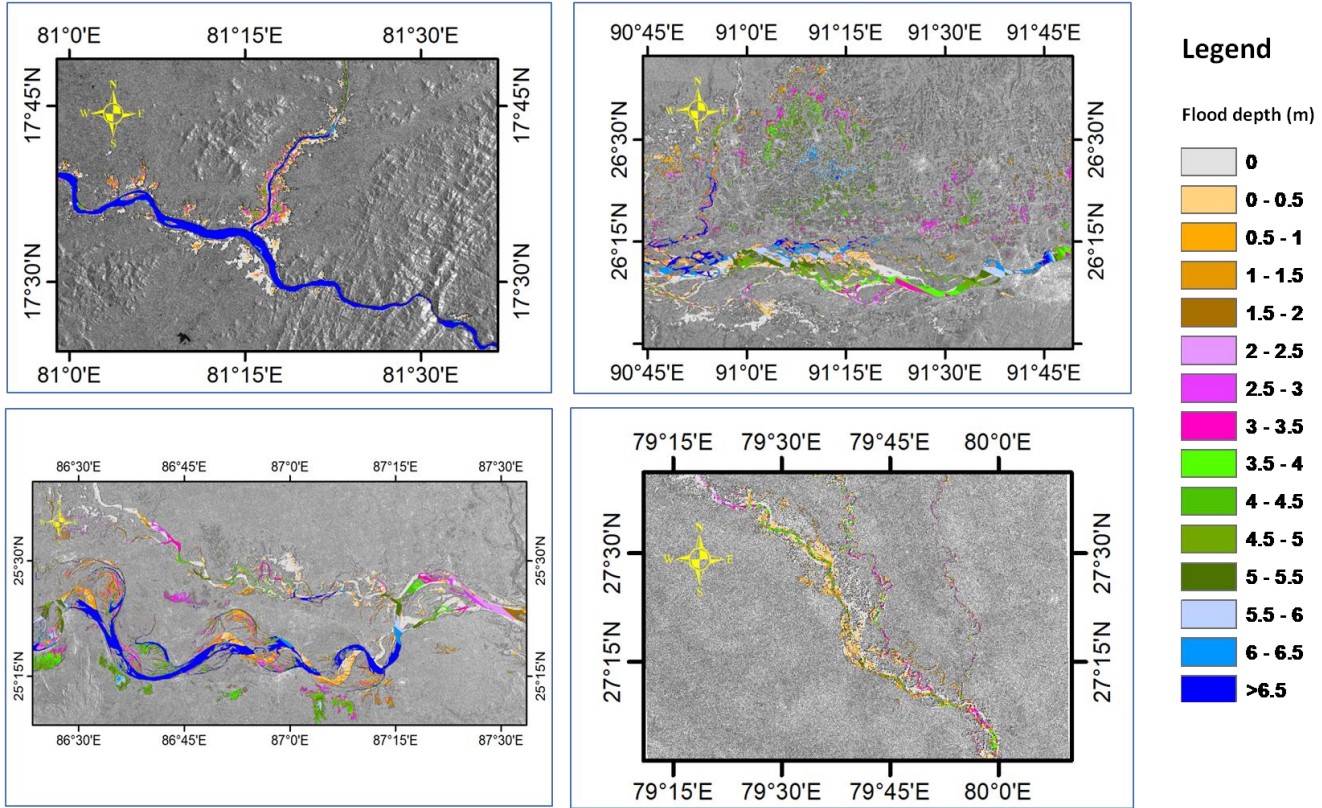

**Figure 8** Flood depths calibrated using TSA technique for (a) Andhra Pradesh (b) Assam (c) Bihar and (d) Uttar Pradesh States

### 5.2.2.2 Validation of Flood depth results

The water levels derived using the Trend Surface Analysis (TSA) technique in four case study areas are compared with field-based water level measurements from gauge stations provided by the Central Water Commission (CWC) for the same date and time. Figure 9 illustrates the method used to compare the TSA-derived water levels and the field measurements.

At each CWC river gauge station, TSA-interpolated water levels were computed. The field-measured water levels at the corresponding location, date, and time served as reference points for the comparison study. Table 7 presents the comparison results and also includes a comparison against water levels estimated using the Floodwater Depth Estimation (FwDET) method. The FwDET water level estimations were performed in the open-source QGIS environment, using the same study area's inundation water layer and a Digital Elevation Model (DEM) as inputs.

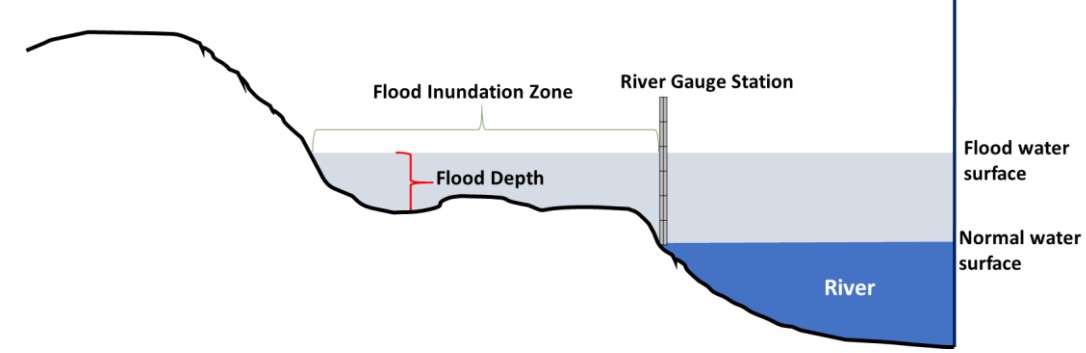

**Figure 9** Pictorial representation of flood plain and river gauge station

**Table 7** Comparison study of water levels among between field measurements, TSA and FwDET

| S.No | Water Gauge Station Name | Field Measured Water Levels | TSA Interpolated Water Levels in meters | FwDET Interpolated Water Levels in meters |
|---|---|---|---|---|
| ANDHRA PRADESH | | | | |
| 1. | Kunavaram | 41.02 | 40.63 | 46.62 |
| 2. | Koida | 39.72 | 39.68 | 42.19 |
| ASSAM | | | | |
| 1. | Beki Rd Bridge | 44.92 | 46.4 | 41 |
| 2. | Pangladiya NT Road Xing | 52.84 | 51.5 | 50.5 |
| 3. | Pandu | 47.25 | 47.12 | 41.5 |
| 4. | Guwahathi | 48.19 | 48.6 | 42 |
| BIHAR | | | | |
| 1. | Baltara | 34.9 | 34.08 | 32.85 |
| 2. | Kahalgaon | 31.08 | 31.459 | 24 |
| 3. | Azamabad | 30.54 | 30.16 | 24 |
| 4. | Kursela | 29.98 | 28.98 | 27 |
| UTTAR PRADESH | | | | |
| 1. | Dabri | 137.18 | 138.6 | 136.21 |
| 2. | Fathegarh | 137.78 | 137.4 | 136.05 |
| 3. | Kannauj | 125.67 | 126 | 125.82 |
| 4. | Bewar | 138.32 | 139.04 | 150.3 |

The results of the floodwater surface derived from surface trend analysis and the Floodwater Depth Estimation Tool (FwDET) indicate that the water surface from the trend analysis closely matches the water surface measurement at CWC gauge stations, whereas the surface derived from the FwDET tool shows significant deviations. TSA estimates deviate from field-level floodwater depth measurements by less than 65cm on average across 14 gauge stations. Most interpolated water levels show a small difference (less than 50cm) compared to field measurements. The underestimation of water levels by the TSA method is primarily due to the presence of real-time gauge stations in upstream flood plains. Conversely, overestimation occurs in areas where gauge stations are located in downstream flood plains.

Trend surface method provides a more balanced and accurate representation of flood surfaces in such cases. However, it is observed that the slope of the flood-affected area plays a significant role in the accuracy of flood depth estimation. For gentle slopes, the accuracy of the TSA method is notably higher. Graphs are plotted as per Figure 10 for the case studies against River gauge station water level and Field measurement, TSA and FwDET methods.

In all case studies, the Trend Surface Analysis (TSA) method outperforms the FwDET method when compared to field measurements. The Root Mean Square Error (RMSE) was calculated for both techniques, with TSA yielding an RMSE of 0.805, whereas FwDET produced an RMSE of 5.23. FwDET estimates generally exhibit sharp transitions in flood depth, while TSA provides a smoother depth distribution. Since TSA-estimated depths also depend on the accuracy of flood extent mapping, the results indicate that the flood maps generated through the automatic tile-based segmentation method appear to be accurate. The turnaround time for this entire process i.e., Flood mapping and Flood depth has taken around 2 min to 5 min depending on the area of the case study on a desktop computer 3.2 GHz processor and 128 GB RAM.

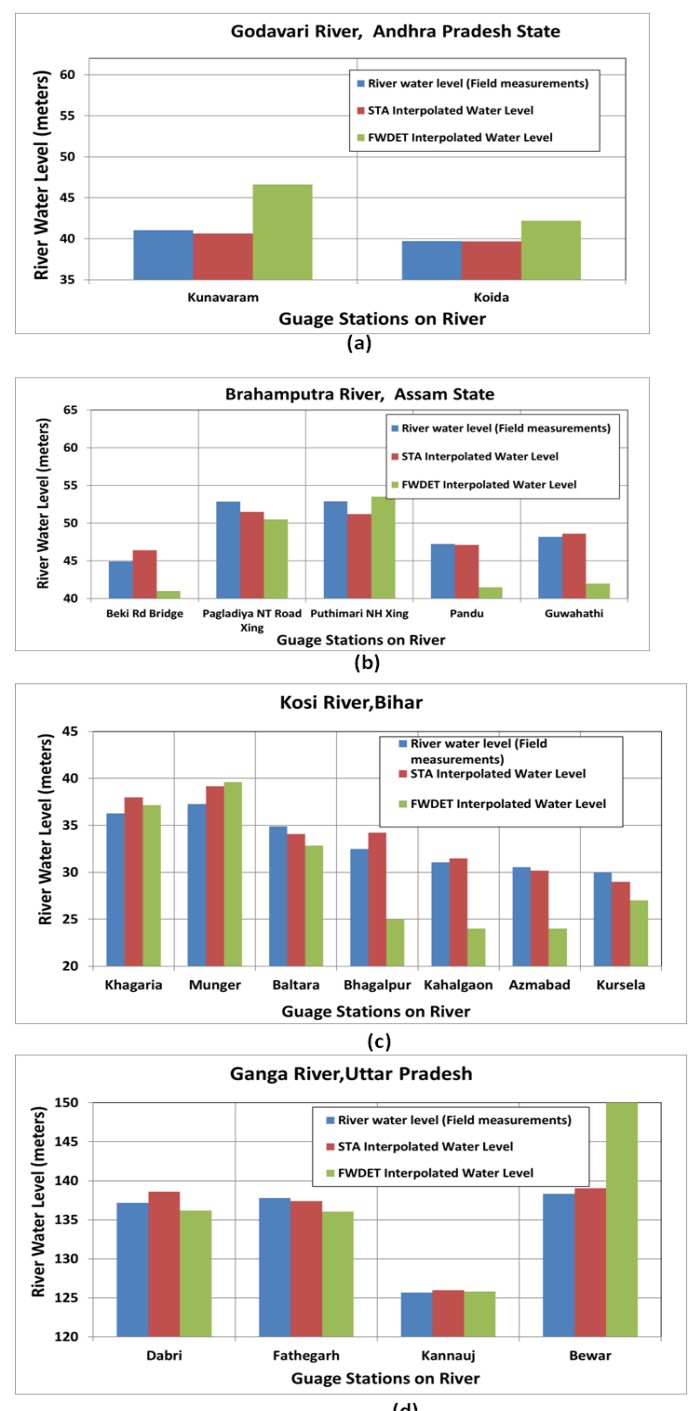

395

**Figure 10** Comparison plots for water levels among field measured data, TSA and FwDET on all study areas

## 5.3 Discussion

This study introduces a novel approach of framework for the rapid estimation of flood extent and depth using data from the EOS-04 satellite, marking the first integration of such a methodology. The proposed method enhances the process of deriving flood extent compared to the automatic tile-based threshold technique proposed by Martinis (2015). While the tiling and tile selection criteria remain consistent with the earlier approach, the application of the thresholding technique is a key differentiator. Specifically, we employed the Otsu thresholding method in this paper instead of the minimum error thresholding method proposed by Kittler and Illingworth (1986). The minimum error thresholding method determines an optimal threshold by modeling pixel intensity distributions as two Gaussian distributions and minimizing the probability of classification error. This method is effective for images where pixel intensities follow a Gaussian distribution, it may not always be optimal where intensity distributions are complex. In contrast, Otsu's method, which maximizes between-class variance, is computationally efficient and more robust for segmenting flood extent in varying conditions. Thus, Otsu's approach was preferred in this study for its reliability in handling diverse flood scenarios. To further improve accuracy, we integrated the Height above the Nearest Drainage (HAND) tool using Digital Elevation Models (DEMs) to refine flood extent estimations. This enhancement addresses limitations in the earlier technique, offering a more topographically accurate representation of flood extent. Unlike Martinis' reliance on TerraSAR-X data operating in the X band, our approach utilizes data from the EOS-04 satellite, which operates in the C band, demonstrating its applicability to a different radar frequency domain. However, the method has certain limitations like high moisture areas are occasionally misinterpreted as flooded regions due to the radar's sensitivity to water content in soil. Furthermore, the method achieves optimal performance when the entire satellite scene is covered under the tile-fitting framework, ensuring comprehensive data representation.

The study also compares the Trend Surface Analysis (TSA) method, a Global Fit interpolation technique, with Local Fit methods like the Floodwater Depth Estimation Tool (FwDET). TSA models the entire surface using global slope patterns, producing a smoother flood depth distribution that captures overall terrain slopes and gradients. This contrasts with FwDET, which interpolates at discrete points and often results in sharp depth transitions (Cohen et al., 2019). TSA produces a smoother distribution of flood depths, effectively capturing overall slope direction and gradients as shown in Figure 8. This makes TSA particularly adept at representing gradual terrain changes and mitigating noise or localized deviations, thereby providing a clearer understanding of flood dynamics over gentle slopes. Also, the TSA technique used in this study is not dependent on the watershed but rather on the slope and height of the terrain. The method relies on how water interacts with the landscape based on the terrain's incline, which directly influences the accuracy of flood depth estimation. When evaluating results against real-time field measurements using RMSE, TSA consistently outperforms FwDET across all study areas. The lower RMSE of TSA indicates its superior accuracy in estimating flood depth compared to FwDET. A key advantage of TSA is that its efficiency is not constrained by the size of the study area; rather, flood extent and hydrologically conditioned DEM plays the most significant role in its accuracy. In contrast, FwDET has predominantly been applied to smaller study areas in previous

research, limiting its applicability to large-scale flood events. However, in real-world flood mapping, large flood extents are more common, making the scalability of an estimation method critical.

TSA has demonstrated superior performance in all aspects, providing more reliable flood depth estimations compared to FwDET. Its ability to adapt to different terrains and flood scenarios highlights its potential as a more effective and scalable approach for flood mapping applications.

TSA-derived flood depths were tested on both LiDAR 5m DEM and Copernicus 30m DEM. The results, shown in Figure 7, reveal a close match in depth estimates for areas with gentle slopes, demonstrating that even coarser DEMs, like the Copernicus 30m DEM, can be effectively used for flood depth estimation in regions with gradual terrain. This broadens the applicability of the method to datasets with varying spatial resolutions. However, the methodology is sensitive to DEM resolution and alignment with the flood layer, sometimes necessitating manual adjustments to ensure proper alignment between the DEM and flood extent.

While the method performs well in areas with gentle slopes, it has limitations in steep terrain where TSA may yield unreliable results. The TSA method fails to accurately represent flood depth in permanent water bodies and reservoir backwaters due to the lack of bathymetric data in DEMs, which only capture surface elevations. This can lead to underestimation or misinterpretation of flood depths, particularly in regions affected by backwater effects. Integrating bathymetric data or hydrodynamic models could significantly improve the accuracy of these estimations. Similarly, using hydrologically conditioned DEMs that account for artificial structures like bridges could also enhance accuracy.

A further challenge in validating flood depth estimates lies in the limited availability of field measurements, which are primarily taken near river gauge stations. This restricts validation to areas near main rivers, making it difficult to assess flood depths farther from these regions. Field measurements are also constrained by accessibility issues, limiting the ability to validate the model across the entire floodplain.

Although the methodology does not account for dynamic hydrodynamic characteristics such as flood velocity or temporal variations, the data generated through this approach is highly useful for real-time flood relief and rehabilitation efforts, particularly in rescue operations. The rapid and automated nature of this framework enables near-real-time flood assessment, supporting emergency response teams in deploying resources such as boats and skilled personnel more effectively. End-users can confidently use this tool for planning mitigation strategies, such as floodplain zoning and infrastructure protection, while recognizing its constraints in predicting dynamic flood behaviours etc.

## 6. Conclusions

In summary, the integration of the Automatic Tile-Based Segmentation Method with the HAND (Height above Nearest Drainage) tool, applied to EOS-04 satellite data, has proven to be a highly effective approach for delineating flood layers. This method addresses key challenges, such as mitigating hill shadows and stray pixels in SAR data to eliminate false water classifications. The study also highlights the sensitivity of the using publicly available DEMs, such as Copernicus 30m DEM,

in regions with gentle slopes where high-resolution DEMs are unavailable. However, for steep flood-prone areas, fine-resolution DEMs remain essential to ensure accurate flood depth estimation.

The adoption of Trend Surface Analysis (TSA) for interpolating water level data further enhances the accuracy and reliability of flood depth estimations, particularly in multi-dimensional river models. TSA effectively captures the spatial trends inherent in river systems, offering improved fitting and precise representations of flood surfaces. When combined, the Automatic Tile-Based Segmentation and TSA techniques have demonstrated robustness and accuracy, as validated against field-measured data. Despite some limitations, this methodology enables rapid flood depth estimation across any flooded area using only a flood layer extent and a hydrologically conditioned DEM. The flood layer defines the inundation extent, while the conditioned DEM ensures accurate elevation representation. Without relying on complex hydrodynamic models or extensive field data, this approach allows quick and efficient flood assessment, making it valuable for emergency response and large-scale flood mapping.

Future research will aim to test this methodology across diverse regions of the country to evaluate its broader applicability. Efforts will also focus on refining the approach to better accommodate varying terrain conditions, including steep slopes, and further improving the alignment and sensitivity of DEM-based flood depth estimations. However, there is a way forward to focus on refining TSA to improve flood depth estimation in steep terrains, extending validation efforts using remote sensing data such as UAV-based LiDAR and Sentinel-3 altimetry, and integrating the proposed methodology into real-time flood monitoring systems to enhance disaster response and large-scale flood assessment.

**Code and data availability**

Nil.

**Author contributions**

Lakshmi Amani and Shashivardhan developed this automated tool. Lakshmi Amani and Suresh Babu tested this tool on field data. Lakshmi Amani, Suresh Babu and Shashivardhan contributed to the paper writing. Durga Rao, Srinivas and Prakash Chauhan technically guided and supervised.

**Competing interests**

The authors declare that they have no conflict of interest.

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
