# Peer review of "Automated Rapid Estimation of Flood Depth using Digital Elevation Model and EOS-04 Satellite derived flood inundation"

_EGUsphere, 2024_

## Author Comment (AC1)

[Figure]

[Figure]

[Figure]

[Figure]

Figure.10. Comparison plots for water levels among field measured data, TSA and FWDET on all study areas

---

## Author Response (AR1)

**Reply to Referee Comments-1**

We would like to sincerely thank the referee for their valuable comments and constructive suggestions. The feedback provided has been instrumental in improving the clarity and depth of our manuscript.

What is the novelty of this paper? Several studies have already been conducted on this topic.

Reply: Though there are similar works done, the novelty of this work lies in the development and application of a comprehensive, automated framework for floodwater delineation and depth estimation using EOS-04 (RISAT-1A) SAR data and Digital Elevation Models (DEMs). This is the first study to apply the established Automatic Tile-Based Segmentation method and the Height above the Nearest Drainage (HAND) tool to EOS-04 data for flood extent delineation. Integrating the proposed method along with the Trend Surface Analysis (TSA) method for flood depth estimation eliminates the need for extensive input data required by traditional hydrodynamic models. The TSA method, capable of generating accurate flood surfaces using only inundated water layers and DEMs, is particularly innovative in its ability to adapt to varying spatial trends and elevation changes in large and complex river systems and efficiently perform in highly flood-affected study areas of India. Furthermore, the incorporation of public and fine-resolution DEMs based on terrain type ensures adaptability and precision in both plain and steep areas. The study also addresses common challenges in SAR data analysis, such as hill shadows, by effectively leveraging the HAND tool to eliminate false water areas. Validation of this methodology derived flood depth against field-measured data and comparison with the Floodwater Depth Estimation Tool (FWDET) demonstrates its superior accuracy, with lower RMSE values, highlighting its potential as a robust, efficient, and scalable solution for real-time flood assessment.

The methods are purely used to reconstruct historical floods. However, the manuscript becomes more impactful if it could be associated with storm characteristics like storm magnitude and return periods to generalize it for future applications. These will provide the probable extent of a storm event with a specific return period.

Reply: As noted in the comments, hydrological and hydrodynamic studies often face challenges due to insufficient and unsuitable data across various terrains. Storm and rainfall events can occur in different areas, impacting administrative units with data limitations. The methods in this study focus on reconstructing historical floods through floodwater delineation and depth estimation. Integrating storm characteristics, such as magnitude and return periods, would enhance the framework's ability to predict future flood scenarios. By linking flood extents and depths with diverse storm events, this methodology could provide probabilistic flood maps for specific return periods, improving decision support for disaster preparedness.

Hydrodynamic models are indeed time- and data-intensive but capable of handling the limitations of the approach raised in this manuscript. This approach is probably effective in flood extent and depth estimation but not sure about its skill on other hydrodynamic characteristics of the flood, e.g., flood velocity. Is it possible to include this property in your analysis?

Reply: The presented approach is designed to efficiently estimate flood extent and depth with minimal data requirements, addressing challenges such as SAR data limitations and terrain-specific

**DEM resolution needs. While hydrodynamic models can simulate additional flood characteristics, such as velocity, they require extensive input data and computational resources. Incorporating flood velocity into this framework is theoretically possible but would require additional data, such as flow rates and channel properties, and potentially adapting or integrating simplified hydrodynamic modelling techniques. Future work could explore combining this methodology with complementary tools or datasets to estimate flood velocity and other dynamic characteristics, enhancing its scope. However, the primary focus of the current approach is on rapid flood assessment with practical applicability, prioritizing efficiency and accessibility over the detailed outputs of traditional hydrodynamic models.**

It is encouraged to specify the limitations of your results in real-world applications and indicate what kind of flood management decisions can be made confidently. This will provide confidence to end-users.

**Reply: The presented approach has some limitations in real-world applications. One of the key challenges of the flood extent method is its occasional misinterpretation of high-moisture areas as flooded regions due to the radar's sensitivity to water content in soil. Additionally, the method performs optimally when the entire satellite scene is captured under the tile-fitting framework, ensuring comprehensive data representation. Similar limitations exist when estimating flood depth. First, the accuracy of flood depth estimation is sensitive to the resolution and alignment of the DEM with the flood layer, particularly in steep terrain, where high-resolution DEMs are essential which is taken care during the selection of satellite datasets and derived flood layers. Second, while the methodology effectively delineates flood extent and depth, it does not account for hydrodynamic characteristics such as flood velocity or temporal variations in flood behaviour, however, the information generated through the proposed approach is of great help in real time relief and rehabilitation, rescue operations in the field. Third, the approach performs best in areas with gentle slopes and may other complex terrains can be handled with slope / land use information in a contextual referencing approach. Despite these limitations, the results are highly reliable for flood extent mapping and depth estimation in plain and moderately sloped regions, enabling decisions such as identifying flood-prone areas, prioritizing evacuation zones, and planning resource allocation for relief and rehabilitation. The rapid and automated nature of the framework makes it suitable for near real-time flood assessment, supporting emergency response efforts. The management decisions, especially during the relief and rehabilitation activities and rescue operations can be made efficiently in terms of deployments of rescue materials like boats/ type boats, and suitably skilled manpower, End-users can confidently use this tool for planning mitigation strategies, such as floodplain zoning and infrastructure protection, while recognizing its constraints in predicting dynamic flood behaviours etc**

In section 4 of the manuscript, results are presented but the discussion part is missing, which is important to connect your results with similar previous studies.

**Reply: Thank you for feedback. We will update in the revised Manuscript.**

**Minor Comments:**

Indicate permanent and seasonal water bodies in your study area (if any)

**Reply:** **In our study area, we have identified both permanent and seasonal water bodies. These features have been incorporated into our analysis, as they can influence flood behaviour and hydrological patterns. We will ensure to clearly indicate these water bodies in the revised manuscript.**

Improve the quality of Figure 10, legends are not readable

**Reply:** **Corrected in the Manuscript and will be updated through revised manuscript.**

Put table titles consistently before/after the table

**Reply:** **Corrected in the Manuscript and will be updated through revised manuscript.**

Correct grammatical and punctuation errors (missing spaces between words, inappropriate use of full stops, colon, etc.)

**Reply:** **Corrected in the Manuscript and will be updated through revised manuscript.**

**Reply to Referee Comments-2**

The paper presents a method for estimating the depth and extent of floods from SAR imagery. Since SAR data are already widely used in the literature for flood monitoring, it would be helpful for the authors to highlight the differences between their approach and existing ones, clearly highlighting the strengths and any limitations. In addition, it is suggested that the authors describe the method in greater detail and clarity so that it can be easily understood and used by a wider audience. The following are additional specific suggestions for improving the quality of the work:

**Reply:** **Thank you for your valuable feedback. We appreciate your suggestion to provide a clearer comparison between our method for estimating flood depth and extent from SAR imagery and existing approaches. We also acknowledge the need to describe our method in greater detail and clarity to ensure it is easily understood and usable by a wider audience. It will be updated in the revised manuscript.**

Line 16: It would be good to make explicit what is meant by the term "lower" and also include numerical performance results for clarity. This would help to better understand the effectiveness of the proposed method.

**Reply:** **The numerical performance results are presented in Section 4.2.3 (Validation of Results), where the RMSE for the Trend Surface Analysis (TSA) technique is reported as 0.805, compared to 5.23 for Flood Water Depth Estimation Tool (FwDET) Technique. To enhance clarity and address your suggestion, abstract from line 15 is improved like this "Water levels estimated at river gauge stations using the TSA technique are validated against real-time field measurements and compared with results derived from the Floodwater Depth Estimation Tool (FwDET). When evaluated relative to gauge station water levels, the TSA technique demonstrates a root mean square error (RMSE) of 0.805, significantly lower than the RMSE of 5.23 observed for the FwDET".**

Lines 40-49: It is recommended to revise the text and punctuation in these lines in order to improve fluency and clarity. Some sentences are indeed a bit complex and could benefit from restructuring.

**Reply: Thank you for valuable suggestion. We have addressed your comments and will update in manuscript. The revised text now reads as follows (lines 40-49): "Additionally, cyclone-prone states such as Odisha, Andhra Pradesh, West Bengal, and Gujarat have necessitated the preparation of Flood Hazard Zonation Atlases, collectively accounting for 10 million hectares of flood-affected areas. This highlights the critical need for real-time flood mapping and monitoring, the implementation of automated flood mapping techniques, and the generation of accurate spatial flood depth information to support disaster management efforts in these regions.**

**Satellite data and flood inundation information are widely used for near real-time mapping and monitoring of flood events (Rizwan Sadiq et al., 2022). Ensuring accuracy in flood extent and depth is critical, as this information is essential for effective relief and rehabilitation efforts in the field."**

Lines 110-111: It is necessary to better specify what is meant by "limit within the active channel." This should be clarified to avoid ambiguity and allow a more precise understanding of the method.

**Reply: Thank you for the suggestion. Statement in lines 110-111, 'Additionally, FwDET's floodwater depth accuracy is poor in the case of active channels,' has been removed to avoid confusion during reading. However, the following line, 'To overcome this limitation, this paper introduces a novel method called Trend Surface Analysis (TSA) to improve the accuracy of flood depth estimation,' is intended to emphasize the novelty of the Trend Surface Analysis (TSA) method in enhancing flood depth estimation accuracy."**

Line 117: The case studies should be described in more detail, including information such as the size of the watersheds and the physical and hydrological characteristics of each. In addition, it would be helpful to add a picture showing the watersheds in relation to the closure sections to enhance visual understanding of the context.

**Reply: Thank you for the feedback. The TSA technique used in this study is not dependent on the watershed but rather on the slope and height of the terrain. The method relies on how water interacts with the landscape based on the terrain's incline, which directly influences the accuracy of flood depth estimation.**

Line 133: It is important to explain the reason why satellite images with different spatial resolution (e.g., CRS and MRS) were used. Also, it would be helpful to clarify what the temporal resolution of acquisition of these images is, especially in relation to the five types of spatial resolution used.

**Reply: Satellite images from multiple sensors are acquired based on their orbital coverage over the study area during the flood event. To ensure higher observation frequency, data from CRS/MRS sensors is utilized when available. The layers selected from these sensors are independent of the temporal resolution.**

Line 142: It would be helpful to know how many level measurements were extracted from the CWC site. It is suggested that these measurements be reported in a graph or table for clearer and more immediate visualization.

**Reply:** **In this study, we employ a single water level measurement from each CWC river gauge site, corresponding to the exact date and time of the satellite acquisition for the study area. For instance, in the Andhra Pradesh study area, satellite imagery was captured on 28th July 2023 at 18:00 hrs. At this precise time, the CWC team recorded the water level measurements for the relevant gauge sites in the study area. A table has been created as per suggestion, and it will also be updated in the manuscript.**

| S.No | Water Gauge Station Name | Field Measured Water Levels |
|------|--------------------------|-----------------------------|
| ANDHRA PRADESH | | |
| 1. | Kunavaram | 41.02 |
| 2. | Koida | 39.72 |
| ASSAM | | |
| 1. | Beki Rd Bridge | 44.92 |
| 2. | Pangladiya NT Road Xing | 52.84 |
| 3. | Pandu | 47.25 |
| 4. | Guwahathi | 48.19 |
| BIHAR | | |
| 1. | Baltara | 34.9 |
| 2. | Kahalgaon | 31.08 |
| 3. | Azamabad | 30.54 |
| 4. | Kursela | 29.98 |
| UTTAR PRADESH | | |
| 1. | Dabri | 137.18 |
| 2. | Fathegarh | 137.78 |
| 3. | Kannauj | 125.67 |
| 4. | Bewar | 138.32 |

Lines 155-185: The authors should explain in more detail the workflow illustrated in Figure 3. In particular, it would be useful to supplement the figure with a textual description that would allow even readers who are not experts in the field to understand the methodological choices made, as well as how the process was replicated.

**Reply:** **We appreciate the reviewer's suggestion to provide a more detailed explanation of the workflow presented in Figure 3. We will update in the revised manuscript.**

Line 274: It is recommended that the Landsat images used to validate the method be introduced in the paragraph devoted to the data used. This would allow for better contextualization of the data and their use in the validation process.

**Reply:** **Thank you for your helpful suggestion. We will include details in the revised manuscript about the source of the Landsat images, their relevance to the study, and how they were used in the validation to ensure better clarity.**

Line 275: It would be appropriate to run the validation on a larger number of dates and create a confusion matrix comparing water and non-water areas. This would allow for a more accurate assessment of method performance. In addition, it would be useful to calculate other performance metrics such as accuracy, precision, and recall to provide a more complete evaluation.

**Reply:** **Thank you for your valuable feedback. The primary focus of this paper is on flood depth estimation. As such, the automatic tile-based segmentation method is not the central point of this study.**

Discussion: The discussion section lacks a comparison with other work in the literature. Authors should highlight the strengths of their method compared to what has been proposed before, pointing out any significant innovations or improvements.

**Reply:** **Thank you for feedback. We will update in the revised Manuscript.**

---

## Referee Report (RR1)

Dear Authors,

Thank you so much for replying to previous comments and uploading the revised version of the manuscript. I am grateful for your hard work and dedication in responding to all the comments raised. However, there are still some major issues that need to be critically addressed. Please find the following major and minor comments that should be taken into account with detailed explanations.

Major Comments:

i. Thank you for explaining the limitations of this method raised in comments #2 and #3. Please indicate where you modified your manuscript (indicate line numbers).

ii. I am still not convinced about the discussion part of this manuscript. I expect more consultation on the previous research findings here. See the presentation quality of similar studies, e.g., https://doi.org/10.1029/2022WR032031.

iii. I couldn't find any suggestions for future research in the manuscript

iv. Please explain the scale of applicability of your methodology with a detailed explanation of why it is limited to that specified scale (if limited to some small or large scale).

v. Please explain how you deal with backwater effects in the complex floodplain. Both HAND and TSA can't consider the effect of backwaters due to artificial structures like canals, bridges, road embankments, levees, etc. Because your DEM never accounts for such infrastructures. Using recently developed high-resolution DEM can help to resolve these problems. However, it might be computationally expensive if you have a large study area. One useful suggestion for this is to manually superimpose these artificial structures on your existing DEM and continue the analysis.

vi. Most of the measurements are only taken from/near the main rivers, which are not representative of the complex floodplains. The main rivers are less dynamic than the floodplain in terms of water depth. Probably, you may not have measurements other than these locations, however, you can use indirect methods like interviewing the local community about the water levels of those flood events with reference to flood marks on known fixed objects/locations like permanent trees, buildings, etc. Please provide limitations if you cannot do this due to specific reasons.

vii. On line 235, you mentioned as you implemented a first-degree polynomial TSA. In my view, the first-degree polynomial is a big simplification for a natural flood plain with diverse topography. The first-degree polynomial assumes a smoothly sloped topography which is way far from your case. Please justify why you chose a first-degree polynomial for TSA. For example, why not a higher-degree polynomial which is more representative?

Minor comments

i. Check subtitle **3.1** on line 140, I believe either you forgot the conjunction word 'and' or have a typo.

ii. Please check editorials like missing spaces (lines 46, 114, 252, 264, 265, 270, 283, 286, 346)

iii. Check for grammatical sentence restructuring (line 275)

iv. Some readers might not be familiar with ERDAS Imagine software (line 288), better to cite a reference for this.

v. Figure 8(e) and its description from lines (346 – 347) is not needed.

vi. Please be consistent with cross-referencing figures e.g., line 289 (Fig 6), line 344 (fig 8), line 375 (fig 10)

vii. The visual quality of Figure 10 is still blurred

---

## Author Response (AR2)

**REFEREE-1 COMMENTS**

Dear Authors, thank you so much for replying to previous comments and uploading the revised version of the manuscript. I am grateful for your hard work and dedication in responding to all the comments raised. However, there are still some major issues that need to be critically addressed. Please find the following major and minor comments that should be taken into account with detailed explanations.

**Thank you for your valuable feedback and for taking the time to review our revised manuscript. We sincerely appreciate your constructive comments and detailed explanations, which have helped us further improve the quality of our work. We have carefully addressed the major and minor issues raised and have submitted a revised version of the manuscript accordingly.**

Major Comments:

i.        Thank you for explaining the limitations of this method raised in comments #2 and #3. Please indicate where you modified your manuscript (indicate line numbers).

**We have incorporated the necessary modifications in the revised manuscript. Regarding the comment on Line 117 earlier, we have included the clarification in the revised manuscript at Line 417: "Also, the TSA technique used in this study is not dependent on the watershed but rather on the slope and height of the terrain. The method relies on how water interacts with the landscape based on the terrain's incline, which directly influences the accuracy of flood depth estimation." Please let us know if any further revisions are needed.**

ii.        I am still not convinced about the discussion part of this manuscript. I expect more consultation on the previous research findings here. See the presentation quality of similar studies, e.g., https://doi.org/10.1029/2022WR032031.

**Thank you for your feedback on the discussion section. We have carefully revised the discussion and updated in manuscript**

iii.        I couldn't find any suggestions for future research in the manuscript

**Thank you for your feedback. We have now included suggestions for future research in the revised manuscript (Lines 461–466), as follows:**

**"Future research will aim to test this methodology across diverse regions of the country to evaluate its broader applicability. Additionally, efforts will focus on refining the approach to better accommodate varying terrain conditions, particularly steep slopes, and improving the alignment and sensitivity of DEM-based flood depth estimations. Further advancements will involve enhancing TSA for improved flood depth estimation in complex terrains, extending validation efforts using remote sensing data such as UAV-based LiDAR and Sentinel-3 altimetry, and integrating the proposed methodology into real-time flood monitoring systems to support disaster response and large-scale flood assessment."**

iv.        Please explain the scale of applicability of your methodology with a detailed explanation of why it is limited to that specified scale (if limited to some small or large scale).

**Thank you for your feedback. We have clarified the scale of applicability in the revised manuscript. Our methodology is not restricted to a specific study area but is more suited for large-scale flood assessments due to the availability of field measurements for validation. Since we compare our results with real-world field measurements from CWC gauge stations, which are typically distributed over large river basins, applying this method to smaller areas is challenging due to the lack of such validation data. However, the approach itself remains scalable and adaptable to different regions, provided that appropriate validation datasets (e.g., UAV-based LiDAR, additional hydrological data) are available. Please let us know if further clarification is needed.**

v.       Please explain how you deal with backwater effects in the complex floodplain. Both HAND and TSA can't consider the effect of backwaters due to artificial structures like canals, bridges, road embankments, levees, etc. Because your DEM never accounts for such infrastructures. Using recently developed high-resolution DEM can help to resolve these problems. However, it might be computationally expensive if you have a large study area. One useful suggestion for this is to manually superimpose these artificial structures on your existing DEM and continue the analysis.

**Thank you for your valuable suggestion. We acknowledge this limitation and have included a discussion in the revised manuscript at Line 436:**

**"The TSA method fails to accurately represent flood depth in permanent water bodies and reservoir backwaters due to the lack of bathymetric data in DEMs, which only capture surface elevations. This can lead to underestimation or misinterpretation of flood depths, particularly in regions affected by backwater effects. Integrating bathymetric data or hydrodynamic models could significantly improve the accuracy of these estimations. Similarly, using hydrologically conditioned DEMs that account for artificial structures like bridges could also enhance accuracy."**

vi.       Most of the measurements are only taken from/near the main rivers, which are not representative of the complex floodplains. The main rivers are less dynamic than the floodplain in terms of water depth. Probably, you may not have measurements other than these locations, however, you can use indirect methods like interviewing the local community about the water levels of those flood events with reference to flood marks on known fixed objects/locations like permanent trees, buildings, etc. Please provide limitations if you cannot do this due to specific reasons.

**Thank you for your insightful comment. We acknowledge this limitation and have addressed it in the revised manuscript at Line 442**

**"A further challenge in validating flood depth estimates lies in the limited availability of field measurements, which are primarily taken near river gauge stations. This restricts validation to areas near main rivers, making it difficult to assess flood depths farther from these regions. Field measurements are also constrained by accessibility issues, limiting the ability to validate the model across the entire floodplain."**

vii.       On line 235, you mentioned as you implemented a first-degree polynomial TSA. In my view, the first-degree polynomial is a big simplification for a natural flood plain with diverse

topography. The first-degree polynomial assumes a smoothly sloped topography which is way far from your case. Please justify why you chose a first-degree polynomial for TSA. For example, why not a higher-degree polynomial which is more representative?

**Thank you for your insightful comment. We acknowledge the concern regarding the selection of a first-degree polynomial for TSA. In the revised manuscript (Line 235), we have clarified our justification: "With a gentle slope or remaining nearly flat across large areas, floodwater follows the path of least resistance, gradually filling depressions and expanding outward rather than forming sharp elevation differences. Given this characteristic, using a first-degree polynomial equation in Trend Surface Analysis (TSA) is a rational approach for modelling the floodwater surface. A first-degree polynomial represents a linear trend, which effectively captures the gradual variation in water surface elevation across the flooded area. This ensures that the estimated water surface reflects the actual spread of floodwater rather than introducing artificial discontinuities that would arise if higher-degree polynomials or abrupt elevation changes were assumed."**

Minor comments

i. Check subtitle 3.1 on line 140, I believe either you forgot the conjunction word 'and' or have a typo.

**Thank you for pointing that out. We have corrected the typo and included the conjunction word 'and' in subtitle 3.1 in the revised manuscript.**

ii. Please check editorials like missing spaces (lines 46, 114, 252, 264, 265, 270, 283, 286, 346)

**Thank you. Updated in revised manuscript.**

iii. Check for grammatical sentence restructuring (line 275)

**Thank you. Updated in revised manuscript.**

iv. Some readers might not be familiar with ERDAS Imagine software (line 288), better to cite a reference for this.

**Thank you. Updated in revised manuscript.**

v. Figure 8(e) and its description from lines (346 – 347) is not needed.

**Figure 8(e) is necessary as it serves as the legend for Figures 8(a) to 8(d). Since the legend remains the same for all subfigures, we have retained it for clarity.**

vi. Please be consistent with cross-referencing figures e.g., line 289 (Fig 6), line 344 (fig 8), line 375 (fig 10)

**Thank you and updated in revised manuscript.**

vii. The visual quality of Figure 10 is still blurred

**Thank you and updated in revised manuscript.**

**REFEREE-2 COMMENTS**

It is recommended to review punctuation and spaces in several places in the document. It is recommended to replace the title of the section 'Results and Discussion' with the term 'Results'.

**Thank you and updated in revised manuscript.**

---

## Author Response (AR3)

**Reply to Referee Comments**

We would like to sincerely thank the referee for valuable comments and suggestions for finalisation of the research article.

1) Consistency and clarification of some abbreviations/names, for example, OTSU' versus 'Otsu' should be checked before publication.

Reply:

1. Total document has been verified to comply with the valuable suggestion and modified. The terms like "Otsu"," FwDET","Floodwater","LiDAR" were corrected.
2. Units viz. 5 meter is mentioned as "5m" throughout the document in the text part.
3. Word to word spacings is corrected wherever required.

2) Better to explain and properly cross-reference the 'Kittler and Illingworth minimum error thresholding' in the discussion which is not mentioned anywhere in the document

Reply:

Explained the concept as suggested in line no. 401 to 407.

3) It is more unusual to consider the main figure and its legend as two different entities

Reply:

The figure 8 is replaced considering the legend integrated within the figure.

*Other Minor Improvements:*

- Figure 3 is replaced by changing the word OTSU to Otsu in the flowchart
- Figure 4(c) is improved for better visualisation of numbers on X and Y axis.
- Line 298: added a sentence "Delineated flood pixels are shown in blue colour."
- Author contributions: First name is added instead of short names in the paragraph.
- Font size of the text in the figures 5,7,8,9 is increased for better visualisation

---

## Author Response (AR4)

Section "Author contributions": please use initials for the authors' names.

**Reply**: I have updated in the manuscript like this "LAC and SVRP developed this automated tool. LAC and SBASV tested this tool on field data. LAC, SBASV and SVRP contributed to the paper writing. DRKHV, SK and PC technically guided and supervised."